# Preserving neuromuscular synapses in ALS by stimulating MuSK with a therapeutic agonist antibody

Sarah Cantor[1], Wei Zhang[1], Nicolas Delestrée[2], Leonor Remédio[1], George Z Mentis[2], Steven J Burden[1]*

[1]Molecular Neurobiology Program, Helen L. and Martin S. Kimmel Center for Biology and Medicine at the Skirball Institute of Biomolecular Medicine, NYU Medical School, New York, United States; [2]Center for Motor Neuron Biology and Disease and Departments of Pathology and Cell Biology and Neurology, Columbia University, New York, United States

**Abstract** In amyotrophic lateral sclerosis (ALS) and animal models of ALS, including *SOD1-G93A* mice, disassembly of the neuromuscular synapse precedes motor neuron loss and is sufficient to cause a decline in motor function that culminates in lethal respiratory paralysis. We treated *SOD1-G93A* mice with an agonist antibody to MuSK, a receptor tyrosine kinase essential for maintaining neuromuscular synapses, to determine whether increasing muscle retrograde signaling would slow nerve terminal detachment from muscle. The agonist antibody, delivered after disease onset, slowed muscle denervation, promoting motor neuron survival, improving motor system output, and extending the lifespan of *SOD1-G93A* mice. These findings suggest a novel therapeutic strategy for ALS, using an antibody format with clinical precedence, which targets a pathway essential for maintaining attachment of nerve terminals to muscle.
DOI: https://doi.org/10.7554/eLife.34375.001

*For correspondence:
steve.burden@med.nyu.edu

## Introduction

Amyotrophic lateral sclerosis (ALS) is a neurodegenerative disease that progresses relentlessly from a subtle decline in motor function to lethal respiratory paralysis within a few years of diagnosis (*Pasinelli and Brown, 2006*; *Taylor et al., 2016*). The disease can be familial and caused by dominant mutations in one of several genes, including *SOD1*, *C9orf72*, *TDP43*, and *FUS* (*Taylor et al., 2016*). More commonly, however, the disease is idiopathic.

Although motor neuron cell death is a hallmark feature of ALS, the loss of neuromuscular synapses occurs prior to the loss of motor neurons and is the primary cause of motor paralysis in both familial and sporadic forms of ALS (*Fischer et al., 2004*; *Schaefer et al., 2005*; *Pun et al., 2006*). The detachment of motor nerve terminals and withdrawal of motor axons has received less attention than the later loss of motor neurons, but therapeutic approaches designed to preserve neuromuscular synapses have the potential to maintain motor function, especially during the early phases of disease, and provide benefit to the quality of life for patient and family.

Transgenic mice bearing dominant mutations in the human *SOD1* gene, including *SOD1-G93A* mice, recapitulate the hallmark features of ALS and provide the most thoroughly studied animal model for ALS (*Vinsant et al., 2013a*; *Vinsant et al., 2013b*). Moreover, because detachment of motor nerve terminals is the primary cause for paralysis in *SOD1-G93A* mice, *SOD1-G93A* mice represent a clinically relevant model for ALS.

The signaling pathways that control attachment of motor axon terminals to muscle are only beginning to be understood, but two genes, *Lrp4* and *Musk*, expressed by muscle, play important

**eLife digest** Amyotrophic lateral sclerosis – often shortened to ALS – is a disease that starts with difficulties moving and progresses to paralysis of many muscles, including those used for breathing. The disease is usually lethal, with patients rarely surviving more than a few years after diagnosis. There is no cure or effective treatment for the disease. It begins with the breakdown of the connections, or synapses, between the muscles and the nerve cells that connect with them. After this, the nerve cell itself breaks down. Many therapeutic approaches have focused on attempts to prevent the nerve cells from dying, but few target the initial degeneration of the synapse.

Cantor et al. asked if intervening when the synapse has already begun to break down could slow the progression of the disease in mice with ALS. Their approach involved using an antibody to bind to a receptor protein called MuSK, which plays an important role in maintaining the synapse between muscle and nerve cell. The antibody boosted the receptor's activity, helping to preserve synapses, including those that connect nerve cells to the diaphragm muscle.

The experiments showed that the antibody treatment led to fewer synapses breaking down, and kept more of the nerve cells alive. Healthier connections between the nervous system and the diaphragm improved the function of this muscle. As a result, the mice given the antibody treatment had a slightly extended lifespan, compared with those given no treatment.

The findings suggest a possible new way to develop treatments for ALS, which could be used in combination with other therapies, such as those aimed at improving the health of the nerve cells. Together, this could improve quality of life for the majority of patients with ALS. Similar strategies could be used to develop treatments to preserve synapses in other neurodegenerative diseases, such as Alzheimer's, Parkinson's and Huntington's disease, as well as some kinds of dementia. Preserving synapses early on, before the significant loss of nerve cells, could help to slow the progression of these diseases, improve the patients' quality of life and extend their lifespans too.
DOI: https://doi.org/10.7554/eLife.34375.002

roles. Lrp4, a member of the LDL receptor family, is the muscle receptor for the critical neuronal ligand, Agrin (*Kim et al., 2008*; *Zhang et al., 2008*). Upon binding Agrin, Lrp4 associates with MuSK, a receptor tyrosine kinase, stimulating MuSK and leading to anchoring and enhanced expression of critical postsynaptic proteins, including Lrp4 (*Burden et al., 2013*). Clustered Lrp4 then signals back to motor axons to stimulate their attachment and differentiation (*Yumoto et al., 2012*).

Recessive mutations in *Agrin*, *Lrp4* or *Musk* cause congenital myasthenia, a group of neuromuscular disorders, distinct from ALS, which compromise the structure and function of neuromuscular synapses and lead to muscle weakness and fatigue (*Engel et al., 2015*). Moreover, autoantibodies to Agrin, Lrp4, or MuSK cause myasthenia gravis (MG), which is likewise distinct from ALS (*Gilhus and Verschuuren, 2015*). In MuSK MG, the pathogenic antibodies are usually directed to the first Ig-like domain in MuSK and reduce MuSK phosphorylation by impairing binding between Lrp4 and MuSK (*Huijbers et al., 2013*; *Koneczny et al., 2013*).

Although defects in the MuSK signaling pathway are not associated with ALS, increasing *MuSK* gene expression stabilizes neuromuscular synapses in *SOD1-G93A* mice, reducing the extent of muscle denervation and improving motor function (*Pérez-García and Burden, 2012*). However, these experiments used transgenic mice to modestly increase *MuSK* expression from muscle, beginning during early development, several months prior to disease onset. Therefore, the therapeutic potential of increasing MuSK signaling as a strategy to reduce denervation and improve motor function in patients diagnosed with ALS remained unclear. Here, we sought to determine whether a pharmacological approach to increase MuSK activity in vivo would preserve neuromuscular synapses in *SOD1-G93A* mice when dosing was initiated after disease onset. This type of approach would have substantially improved potential for translation to ALS patients without the complex requirements for gene therapy (*Miyoshi et al., 2017*).

## Results

### Agonist antibodies to MuSK

A previous study identified twenty-one single chain antibodies (scFvs) that recognize mouse MuSK and raised the idea that a subset of these antibodies may function as MuSK agonists in vivo (*Xie et al., 1997*). We studied the activity of two antibodies, #13 and #22, reported to stimulate MuSK in cultured myotubes, as well as antibody #21, reported to bind but not stimulate MuSK. We confirmed that antibodies #13 and #22, re-engineered as human IgG1 molecules, stimulated MuSK tyrosine phosphorylation in the C2 mouse muscle cell line (*Figure 1A*), whereas antibody #21, as well as a control IgG1 antibody to ragweed pollen, failed to stimulate MuSK phosphorylation (*Figure 1A*).

Agrin stimulates MuSK phosphorylation by binding Lrp4, which promotes association between Lrp4 and MuSK, requiring the first of three Ig-like domains in MuSK (*Zhang et al., 2011*). In contrast to the Agrin-dependent mechanism for activating MuSK, the agonist antibody binds the Fz-like domain in MuSK, force-dimerizing and stimulating MuSK phosphorylation, independent of Lrp4 (*Figure 1B,C* and *Figure 1—figure supplement 1*). Importantly, the Fz-like domain is dispensable for synapse formation in mice (*Remédio et al., 2016*).

### MuSK agonist antibodies engage MuSK in vivo

To determine whether agonist antibody #13 could engage MuSK in vivo, we intraperitoneally (IP) injected varying amounts of the MuSK agonist antibody on a human IgG1 backbone, or a control human IgG1 antibody to ragweed pollen, into wild type mice. Several days later, we stained whole mounts of the diaphragm muscle to determine whether the agonist antibody engaged MuSK at the synapse. *Figure 2A* shows that neuromuscular synapses were labeled specifically by the MuSK agonist antibody. MuSK staining was evident as early as 3 days (*Figure 2Aiv-vi*), and staining persisted for at least 7 days after the single injection (*Figure 2A,vii-ix*). The organization of AChRs and nerve terminals appeared normal (*Figure 2A,ii,v,viii*), indicating that the MuSK agonist antibody did not disturb major features of synaptic differentiation. Moreover, visual observation of the antibody-injected mice did not reveal overt behavioral abnormalities, indicating that the MuSK agonist antibody was well tolerated by the mice. Two mg/kg of the agonist antibody was sufficient to saturate MuSK labeling at the synapse (*Figure 2B*) and increase MuSK tyrosine phosphorylation in vivo (*Figure 1—figure supplement 1*).

We measured the pharmacokinetic properties of the injected antibody and found that the half-life of the injected antibody in blood was ~12 days (*Figure 2C*). The antibody exhibited linear clearance for 21 days after antibody injection, indicating that exposure could be maintained over several weeks. In addition, these results demonstrated that the mouse immune system did not recognize and clear the antibody, which contained a human Fc region, from the circulation over this three-week time period (*Figure 2C*).

### Single dose of MuSK agonist antibody decreases denervation in *SOD1-G93A* mice

We studied female and male *SOD1-G93A* mice, on a C57BL/6 background, with 21–26 copies of the human *SOD1-G93A* gene (*Figure 3—figure supplement 1*). In *SOD1-G93A* mice, denervation of limb muscles begins at P50, whereas denervation of the diaphragm muscle begins a month later (*Pun et al., 2006*; *Rocha et al., 2013*). Because denervation of the diaphragm muscle is responsible for lethal respiratory paralysis, we focused our analysis on innervation of this muscle. We first quantified the extent of innervation in the diaphragm muscle at P90 by staining for nerve terminals and postsynaptic AChRs, which remain even at denervated synaptic sites (*Figure 3A*). Denervation was evident in *SOD1-G93A* mice as early as P90 (*Figure 3B,C*). From P90 to P110, the extent of full innervation, defined as perfect apposition of nerve terminals and the AChR-rich postsynaptic membrane, decreased from 77.3% to 18.1% in female and from 53.1% to 16.1% in male *SOD1-G93A* mice (*Figure 3B*). Likewise, the extent of complete denervation increased from 2.3% to 41% in female and from 16.7% to 24.4% in male *SOD1-G93A* mice over this twenty-day period (*Figure 3C*).

*SOD1-G93A* mice were injected with the MuSK agonist antibody at P90. Because the antibody had a half-life of 12 days and 2 mg/kg of antibody saturated MuSK at the synapse (*Figure 2B,C*), we

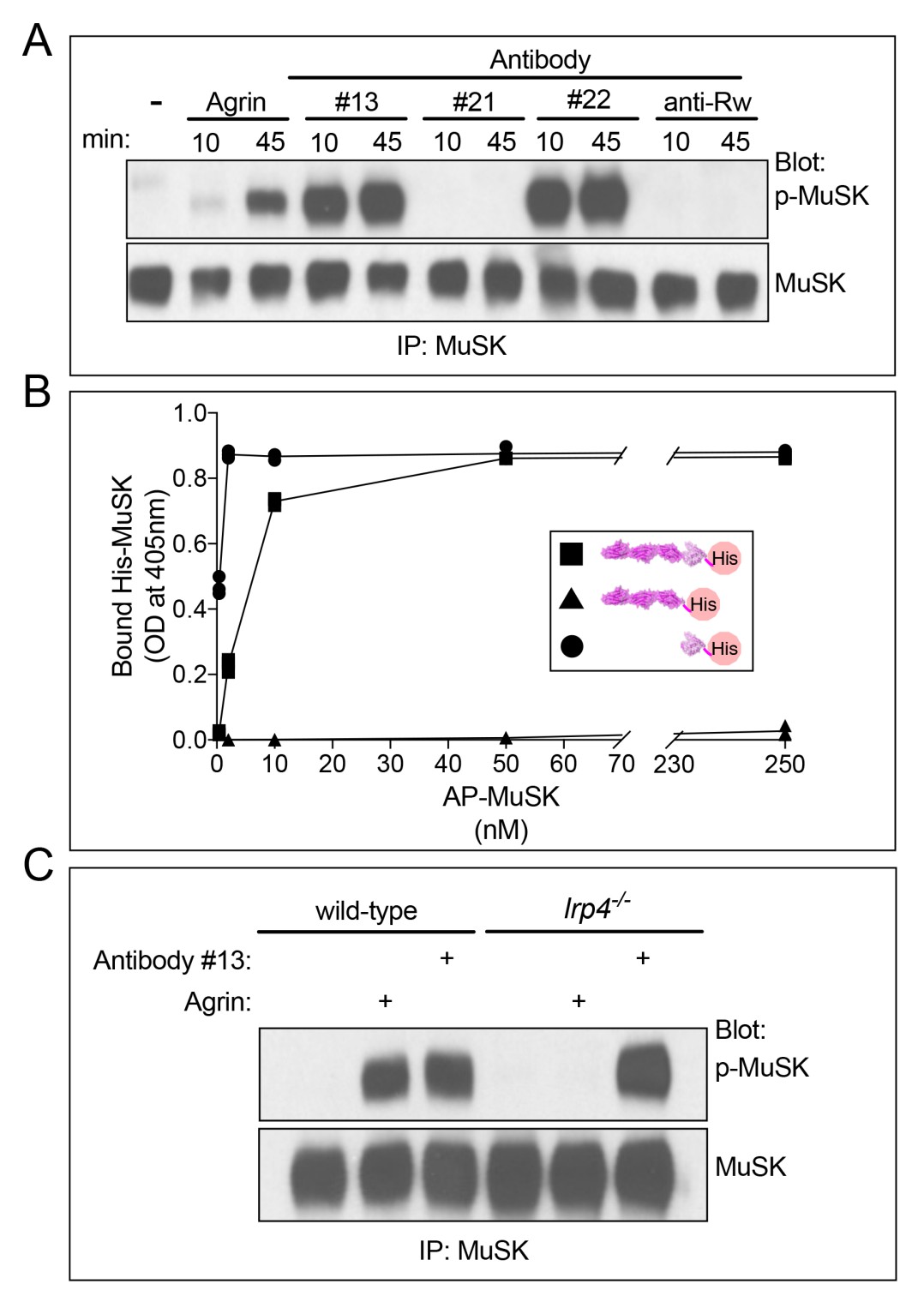

**Figure 1.** MuSK agonist antibodies activate MuSK, independent of Lrp4, by binding the Fz-like domain in MuSK. (**A**) C2 myotubes were treated with neural Agrin or antibodies for the indicated times. MuSK was immunoprecipitated, and Western blots were probed for MuSK or phosphotyrosine. Neural Agrin and MuSK antibodies #13 and #22 stimulate MuSK tyrosine phosphorylation in C2 myotubes, whereas MuSK antibody #21 and a control antibody to Ragweed pollen (Rw) failed to stimulate MuSK phosphorylation. (**B**) We used a solid-phase binding assay to measure binding of His-tagged MuSK proteins to microtiter wells coated with MuSK agonist antibody #13. The scatter plot shows that full-length ecto-MuSK (■), as well as the MuSK Fz-like domain

*Figure 1 continued on next page*

*Figure 1 continued*

alone ( ● ) bind MuSK antibody #13 in a dose-dependent and saturable manner; in contrast, the first three Ig-like domains in MuSK ( ▲ ) fail to bind the MuSK agonist antibody (n = 3). (**C**) Wild type and *Lrp4* mutant myotubes were treated with neural Agrin or MuSK agonist antibody #13. Agrin stimulates MuSK phosphorylation in wild type but not *Lrp4* mutant myotubes, whereas MuSK agonist antibody #13 stimulates MuSK phosphorylation in both wild type and *Lrp4* mutant myotubes.

DOI: https://doi.org/10.7554/eLife.34375.003

The following figure supplement is available for figure 1:

**Figure supplement 1.** MuSK agonist antibody #13 stimulates MuSK tyrosine phosphorylation in vivo.

DOI: https://doi.org/10.7554/eLife.34375.004

injected *SOD1-G93A* mice with 10 mg/kg of agonist antibody, ensuring that the antibody concentration in blood would remain at saturating levels for MuSK-binding over the 20 day period. We found that a single dose of the MuSK agonist antibody increased the number of fully innervated synapses by 2.7- and 2.5-fold in female and male *SOD1-G93A* mice, respectively, and decreased the number of fully denervated synapses by 3.7- and 2.3-fold in female and male *SOD1-G93A* mice, respectively (*Figure 3B,C*). These findings demonstrated that the MuSK agonist antibody, introduced after disease onset, decreased motor axon withdrawal from the diaphragm muscle.

## Chronic dosing with the MuSK agonist antibody halts further denervation in *SOD1-G93A* mice for over two months

To determine whether the MuSK agonist antibody could preserve neuromuscular synapses over a longer time period, we chronically dosed *SOD1-G93A* mice. To avoid host recognition and clearance of the antibody during chronic exposure, we used a MuSK #13 antibody on a murine IgG2a backbone that also lacked effector functions (*Lo et al., 2017*). The ability of this 'reverse chimera' to bind and stimulate MuSK was similar to the antibody with a human IgG backbone (*Figure 4—figure supplement 1*). Moreover, the 'reverse chimera' had a half-life similar to the human agonist antibody in vivo (*Figure 4—figure supplement 2*).

*SOD1-G93A* mice were injected with 10 mg/kg of the reverse chimera agonist antibody at P90 and every 24 days thereafter, and we sacrificed chronically injected mice every 24 days to quantify innervation of the diaphragm muscle (*Figure 4A*). Because 2 mg/kg of antibody saturated MuSK at the synapse and because the antibody had a 11 day half-life in blood, this dosing schedule ensured that saturating levels of the MuSK agonist antibody were maintained at all times (*Figure 4—figure supplement 2*).

In *SOD1-G93A* mice injected with a control antibody to GP120, synaptic loss continued to decline from P114 through P162, so that only 11% of the synapses were fully innervated at P162 (*Figure 4B*). This progressive loss was halted by injection of the MuSK agonist antibody, as the number of fully innervated synapses was largely unchanged (40–50%) from P114 to P162 in *SOD1-G93A* mice injected with the MuSK agonist antibody (*Figure 4B*). Similarly, the number of fully denervated synapses continued to increase from P114 through P162 in *SOD1-G93A* mice injected with the control antibody, whereas this progressive increase was prevented by the MuSK agonist antibody (*Figure 4C*). These findings indicate that the MuSK agonist antibody prevented further synaptic loss and preserved synapses for at least 50 days after signs of denervation and disease were evident in *SOD1-G93A* mice.

During disease progression, synapses transition through a partially innervated phase, when only a portion of the AChR-rich postsynaptic membrane is apposed by motor nerve terminals (*Figure 4—figure supplement 3*). Although the number of partially innervated synapses was similar in *SOD1-G93A* mice injected with the control or MuSK agonist antibody (*Figure 4—figure supplement 3*), the extent of nerve terminal coverage was 34% greater at partially innervated synapses in mice injected with the MuSK agonist antibody (*Figure 4—figure supplement 3*). Thus, the MuSK agonist antibody increased both full innervation as well as nerve terminal coverage at partially innervated synapses in *SOD1-G93A* mice.

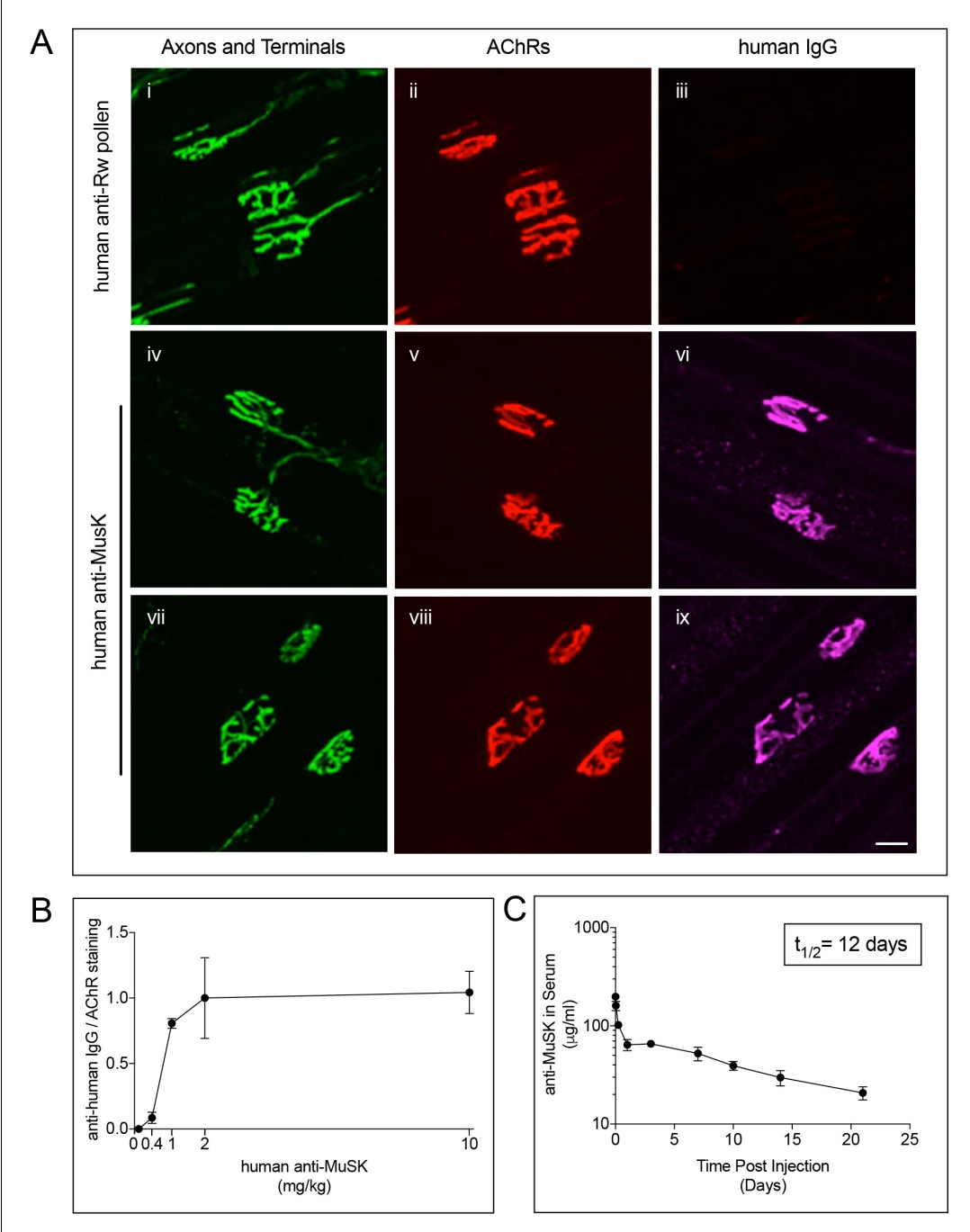

**Figure 2.** MuSK agonist antibody #13 engages MuSK at the synapse shortly after IP injection. (**A**) Staining for the injected MuSK antibody (10 mg/kg) was evident as early as three days (iv-vi) and persisted for at least seven days after a single injection of antibody (vii-ix). A human antibody to Ragweed (Rw) pollen (10 mg/kg) failed to stain synapses (i-iii) (scale bar = 20 μm). (**B**) 2 mg/kg of the injected MuSK agonist antibody saturated MuSK at the synapse. The ratio of MuSK/AChR staining at 2 mg/kg antibody was assigned a value of 1.0 (±SEM, n = 3), and the values at other doses were expressed relative to this value. (**C**) Following a single injection of human MuSK antibody #13 (10mg/kg) the level of antibody in serum declines with a half-life of 12.1 days. n = 3.

DOI: https://doi.org/10.7554/eLife.34375.005

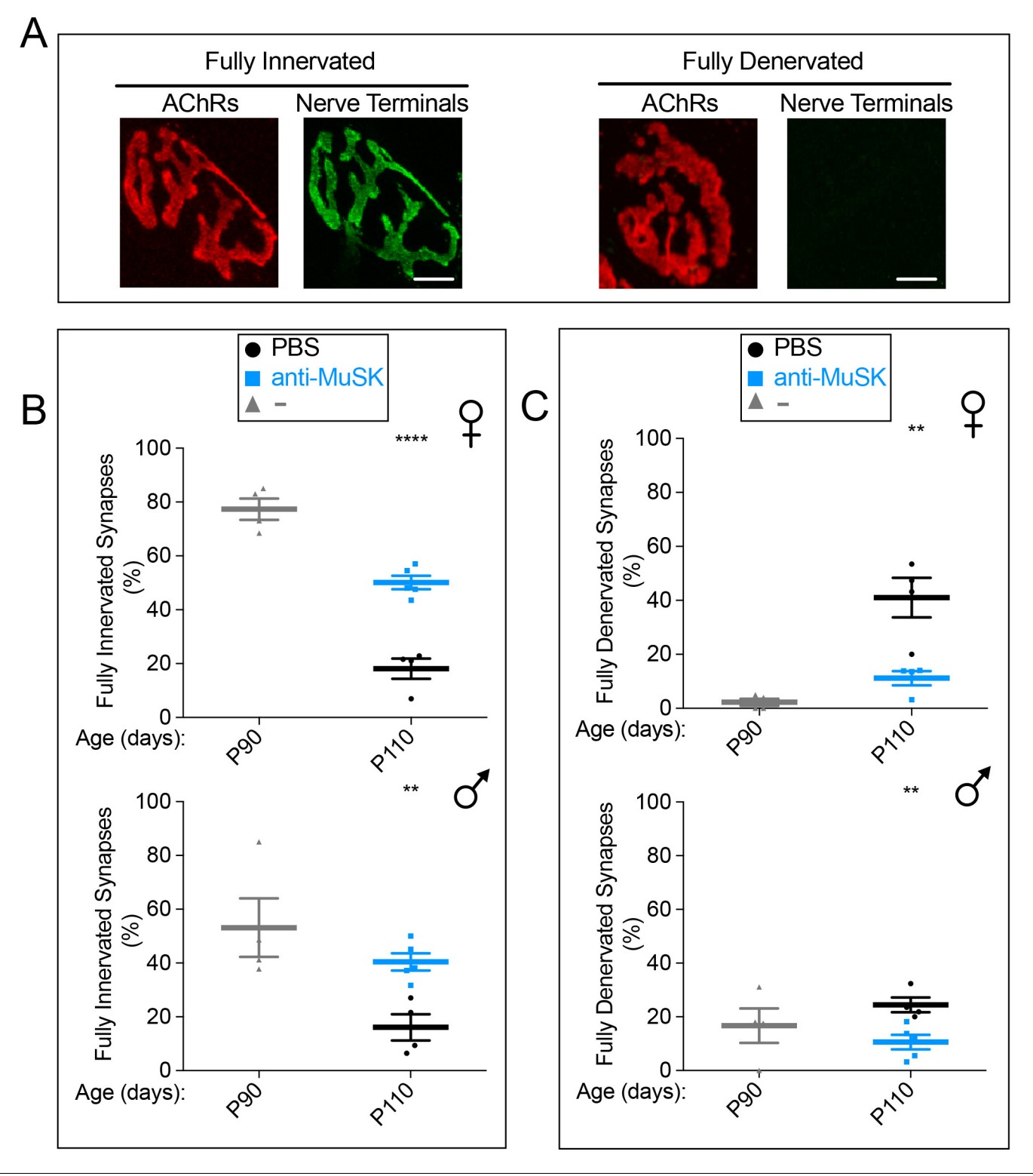

**Figure 3.** A single injection of the MuSK agonist antibody at P90 reduced synaptic loss for twenty days in *SOD1-G93A* mice. (**A**) AChRs are concentrated in the postsynaptic membrane at innervated and fully denervated synapses (scale bar = 20 μm). (**B, C**) Denervation is evident in female and extensive in male *SOD1-G93A* mice at P90 (▲). Over the next twenty days, the extent of full denervation increases and the number of fully innervated synapses decreases (●). A single injection of agonist antibody #13 (■) reduces the extent of denervation and the loss of innervation. The scatter plot shows the values for individual mice (n = 4 or 5), as well as the mean values and SEM; **p<0.01, ****p<0.0001.

*Figure 3 continued on next page*

*Figure 3 continued*

DOI: https://doi.org/10.7554/eLife.34375.006

The following figure supplement is available for figure 3:

**Figure supplement 1.** *hSOD1-G93A* copy number remained unchanged over generations and throughout the experiments.
DOI: https://doi.org/10.7554/eLife.34375.007

## Improved motor system output of the diaphragm muscle

To determine whether maintaining neuromuscular synapses led to improved motor system output, we used an ex-vivo phrenic nerve/diaphragm muscle preparation to measure the compound muscle action potentials (CMAPs), following phrenic nerve stimulation. We studied *SOD1-G93A* mice three to four weeks prior to end-stage (*Figure 5*). We stimulated the phrenic nerve to the diaphragm muscle and recorded CMAPs, which elicit muscle contraction (*Figure 5—figure supplement 1*). We found no significant difference in the amplitude of the first CMAP between *SOD1-G93A* mice injected with the MuSK agonist antibody or the control antibody to GP120 (anti-GP120-treated males: $5.95 \pm 1.14$ mV; anti-MuSK-treated males: $5.93 \pm 0.52$ mV; anti-GP120-treated females: $4.95 \pm 0.54$ mV; anti-MuSK-treated females: $5.81 \pm 0.63$ mV). We next measured the reliability of synaptic transmission at the neuromuscular junction by repetitively stimulating the phrenic nerve at a physiological frequency (20 Hz). We found a rapid and severe decline in the amplitude of the CMAP, indicative of synaptic dysfunction and denervation, in *SOD1-G93A* mice chronically injected with the control antibody to GP120. In contrast, the decline in CMAP amplitude was far less severe in *SOD1-G93A* mice treated with the MuSK agonist antibody, demonstrating that the MuSK agonist antibody improved neuromuscular function (*Figure 5*). Moreover, repetitive stimulation of the phrenic nerve at a more challenging frequency (50 Hz) led to frequent failures to elicit a CMAP in *SOD1-G93A* mice injected with the control antibody to GP120. Such failures were less frequent in *SOD1-G93A* mice injected with the MuSK agonist antibody, similar to wild type mice (*Figure 5*). These CMAP failures are likely due to presynaptic mechanisms, such as conduction block or impaired neurotransmitter release, rather than the inability of motor end plates to generate an action potential. In either case, the maintenance of neuromuscular synapses, stimulated by the MuSK agonist antibody, led to improved reliability of synaptic transmission and output of the critically important diaphragm muscle in *SOD1-G93A* mice.

## MuSK agonist antibody decreases motor neuron loss in *SOD1-G93A* mice

We next assessed whether preserving neuromuscular synapses in *SOD1-G93A* mice reduced motor neuron death. During embryonic development motor neuron death is regulated by innervation and reduced when motor neurons make additional synapses with muscle (*Hollyday and Hamburger, 1976*; *Tanaka and Landmesser, 1986*; *Landmesser, 1992*), whereas survival of adult motor neurons is less dependent upon muscle innervation (*Lowrie and Vrbová, 1992*). We quantified the number of motor neurons, stained for choline acetyltransferase (ChAT), in the lumbar spinal cord of *SOD1-G93A* mice injected chronically either with the control antibody to GP120 or the MuSK agonist antibody (*Figure 6A*). The MuSK agonist antibody increased the number of motor neurons by 31% to 57% at P138 (*Figure 6B*), during the peak period of motor neuron cell death in *SOD1-G93A* mice when approximately half of spinal motor neurons have been lost (*Vinsant et al., 2013a*). These findings demonstrate that increasing retrograde signaling after disease onset not only preserves neuromuscular synapses but also promotes survival of spinal motor neurons in *SOD1-G93A* mice.

## MuSK agonist antibody extends lifespan of *SOD1-G93A* mice

Denervation of the diaphragm muscle is responsible for lethal respiratory paralysis in *SOD1-G93A* mice and ALS. We therefore asked whether maintaining neuromuscular synapses and improving output of the diaphragm muscle extended the lifespan of *SOD1-G93A* mice. Female *SOD1-G93A* mice injected with the control antibody to GP120 had an average lifespan of 169 days (see Materials and methods), whereas male *SOD1-G93A* mice injected with the control antibody had an average lifespan of 157.5 days (*Figure 6C,D*). Chronic injection with the MuSK agonist antibody prolonged survival of female and male *SOD1-G93A* mice by 7 ($p<0.05$) and 10 days ($p<0.001$), respectively

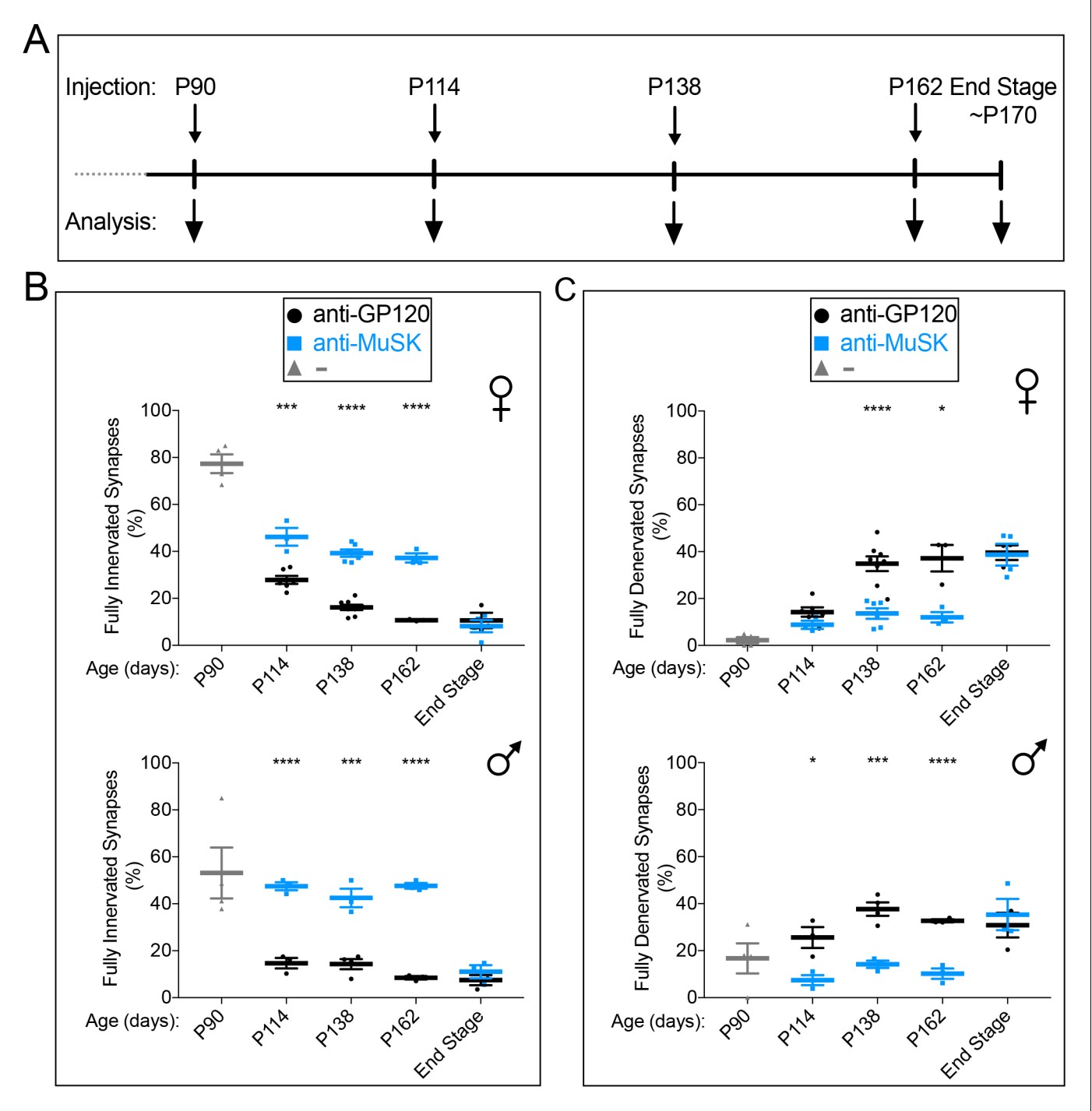

**Figure 4.** Chronic dosing with the MuSK agonist antibody prevents further denervation in *SOD1-G93A* mice. (**A**) The reverse chimera MuSK agonist antibody #13 was injected at P90 ( ▲ ) (**B**) The extent of full innervation decreases progressively from P90 to P162 in female and male *SOD1-G93A* injected with a control antibody to GP120 ( ● ). The reverse chimera MuSK agonist antibody #13 ( ■ ) halts this progressive loss, as the number of fully innervated synapses is unchanged between P114 and P162. (**C**) Full denervation increases progressively from P90 to P162 in female and male *SOD1-G93A* injected with a control antibody to GP120 ( ● ). The reverse chimera MuSK agonist antibody #13 prevents this progressive increase in denervation, as the number of fully denervated synapses is unchanged between P114 and P162 ( ■ ). At disease end-stage, the number of fully innervated and denervated synapses was identical in *SOD1-G93A* mice injected with the MuSK agonist or control antibody. The scatter plot shows the values for individual mice (n = 3 to 8), as well as the mean values and SEM; *p<0.05, ***p<0.001, ****p<0.0001.
DOI: https://doi.org/10.7554/eLife.34375.008

The following figure supplements are available for figure 4:

*Figure 4 continued on next page*

*Figure 4 continued*

**Figure supplement 1.** The human and reverse chimera versions of MuSK agonist antibody #13 induce acetylcholine receptor (AChR) clustering in C2C12 myotubes whereas a Fab from MuSK antibody #13 fails to stimulate AChR clustering.
DOI: https://doi.org/10.7554/eLife.34375.009

**Figure supplement 2.** The reverse chimera MuSK agonist antibody #13 has a half-life of 11 days and chronic dosing with this antibody maintains the agonist antibody at levels that are sufficient to saturate MuSK at the synapse.
DOI: https://doi.org/10.7554/eLife.34375.010

**Figure supplement 3.** Chronic dosing with the MuSK agonist antibody increases the extent of nerve terminal coverage at partially innervated synapses in *SOD1-G93A* mice.
DOI: https://doi.org/10.7554/eLife.34375.011

(*Figure 6C,D*). Thus, the MuSK agonist antibody, introduced after disease onset, slowed the disassembly of neuromuscular synapses, improved motor output of the diaphragm muscle and extended the lifespan of *SOD1-G93A* mice.

## Discussion

ALS is a devastating disease that progresses in a relentless manner from detachment of motor nerve terminals to lethal respiratory paralysis within several years of diagnosis. Currently, there is an unmet need for therapies that significantly alter the course of disease. Here, we describe a therapeutic approach designed to slow the loss of motor innervation to muscle by targeting a well-defined molecule and mechanism for forming and maintaining neuromuscular synapses. We show that an agonist antibody to MuSK, introduced after disease onset, decreases muscle denervation, improves motor system output, reduces motor neuron loss and extends survival in an aggressive mouse model of ALS. If this strategy, described here for an aggressive mouse model of ALS, were similarly successful in preserving innervation in sporadic and familial ALS, this therapeutic approach would have the potential to improve the quality of life for ALS patients, and as such warrants further study.

Anti-sense RNA directed toward *SOD1* is currently being tested as a promising therapeutic for ALS caused by mutations in *SOD1* (*Miller et al., 2013*). A similar approach may ultimately be effective for other dominant, familial forms of ALS (*Reddy and Miller, 2015*; *van Zundert and Brown, 2017*). However, >80% of ALS patients are diagnosed with sporadic ALS, so strategies to inactivate a single culprit gene are not tenable for most cases of ALS. Instead, multiple, concurrent therapeutic interventions that effectively address the pathology and symptoms of ALS will likely be necessary to alter the course of disease (*Brown and Al-Chalabi, 2017*).

Because synaptic loss and muscle denervation are common to sporadic as well as familial forms of ALS, the approach described here has the potential to be effective for both forms of ALS. Moreover, increasing MuSK activity and retrograde signaling may also slow the deterioration of neuromuscular synapses in other neuromuscular diseases and during aging (*Engel et al., 2015*; *Gilhus and Verschuuren, 2015*; *Valdez et al., 2012*; *Poort et al., 2016*). Consistent with this idea, adenoviral expression of Dok-7, an inside-outside activator of MuSK, not only extends longevity of *SOD1-G93A* mice but also provides benefit in other mouse models of neuromuscular disease, including congenital myasthenia and Emery-Dreifuss muscular dystrophy (*Miyoshi et al., 2017*; *Arimura et al., 2014*). Further, there is increasing evidence that synaptic loss occurs early during disease progression in other neurodegenerative diseases, such as Alzheimer's disease, Huntington's disease, Parkinson's disease and Frontotemporal dementia and Spinal Muscular Atrophy (*Henstridge et al., 2016*), so similar strategies, designed to preserve synapses, may slow progression in these diseases as well.

Our proof of concept experiments were designed to determine whether boosting retrograde signaling in vivo with the MuSK agonist antibody might slow motor axon withdrawal and muscle denervation in *SOD1-G93A* mice. As such, we introduced the MuSK agonist antibody after denervation was already evident, during the early phase of denervation in female *SOD1-G93A* mice and mid-phase in male *SOD1-G93A* mice, but before *SOD1-G93A* mice exhibited overt and severe deficits in limb motor function. This timing for delivery of the MuSK agonist antibody may be pertinent and significant for ALS, as denervation is the cause of muscle fibrillations, an early clinical sign in ALS. Because MuSK-dependent retrograde signaling is likely to act focally on nerve terminals and axons that are near the postsynaptic membrane and to be less effective in promoting regeneration of

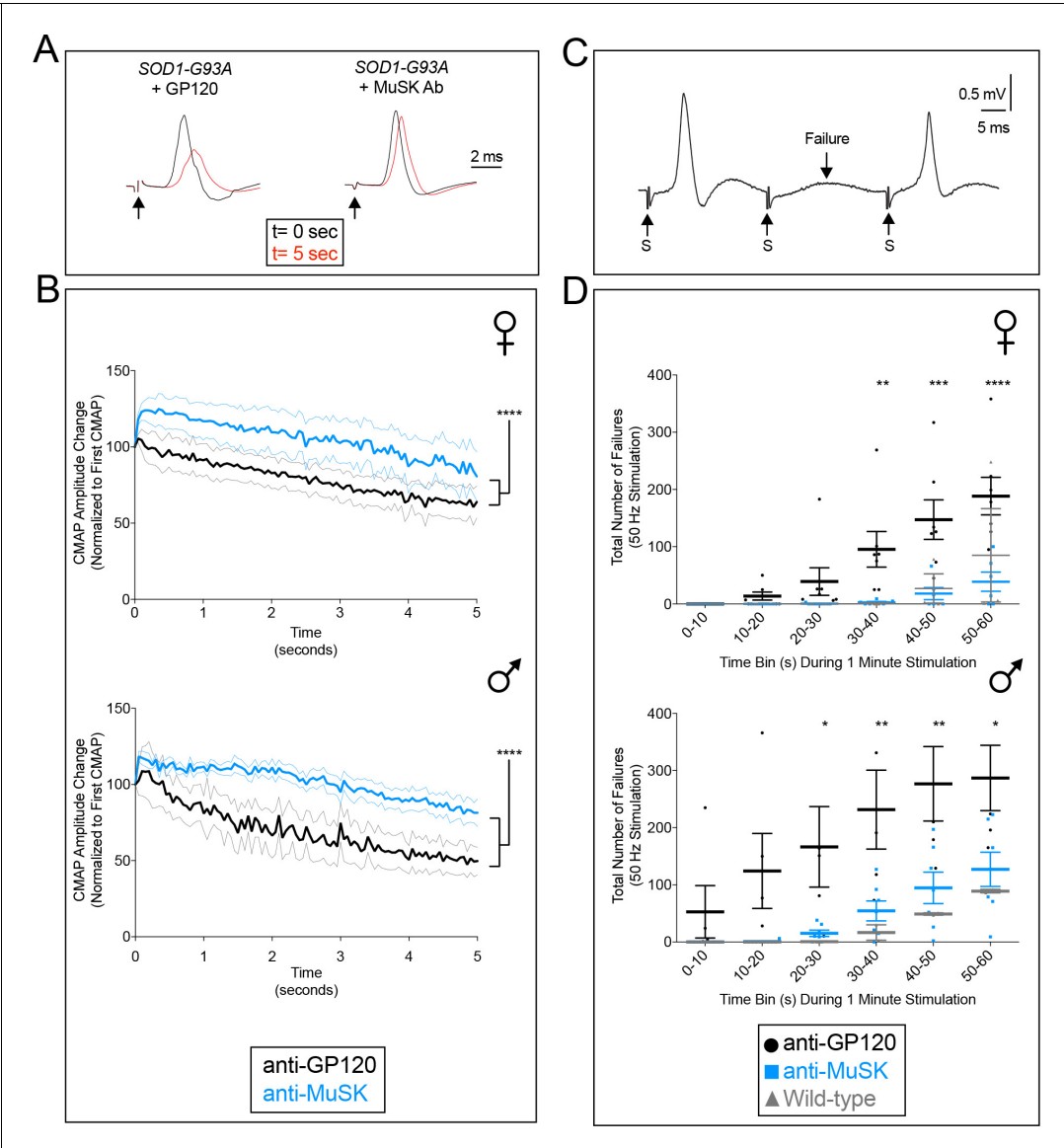

**Figure 5.** The MuSK agonist antibody improves motor system output in the diaphragm muscle. (A,B) 20 Hz stimulation (arrow) of the phrenic nerve from *SOD1-G93A* mice injected with the control antibody to GP120 led to a rapid and severe decline in the CMAP amplitude. In contrast, the CMAP amplitude decreased gradually and modestly in *SOD1-G93A* mice injected with the MuSK agonist antibody. After 5 s, the MuSK agonist improved CMAP amplitude by 13.6% in females and by 31.7% in males (n = 6–7; p<0.0001). The faint grey and blue lines indicate the SEMs. (C,D) 50 Hz stimulation (S, arrow) of the phrenic nerve in *SOD1-G93A* mice injected with the control antibody to GP120 led to frequent failures (F, arrow) to elicit a CMAP, whereas CMAPs were reliably elicited in *SOD1-G93A* mice injected with the MuSK agonist antibody, similar to the number of failures seen in wild type mice. The MuSK agonist antibody reduced the number of failures by 88% in females and 70% in males during 1 min of stimulation. The scatter plot shows the values for individual mice, as well as the mean values and SEM; *p<0.05, **p<0.01, ***p<0.001. The baseline CMAP amplitude data are as follows: anti-GP120-treated males, 5.95 ± 1.14 mV; MuSK agonist antibody-treated males, 5.93 ± 0.52 mV; anti-GP120-treated females, 4.95 ± 0.54 mV; MuSK agonist antibody-treated females, 5.81 ± 0.63 mV.

DOI: https://doi.org/10.7554/eLife.34375.012

The following figure supplement is available for figure 5:

**Figure supplement 1.** Drawing of experimental protocol to stimulate the phrenic nerve and record CMAPs in the diaphragm muscle.

DOI: https://doi.org/10.7554/eLife.34375.013

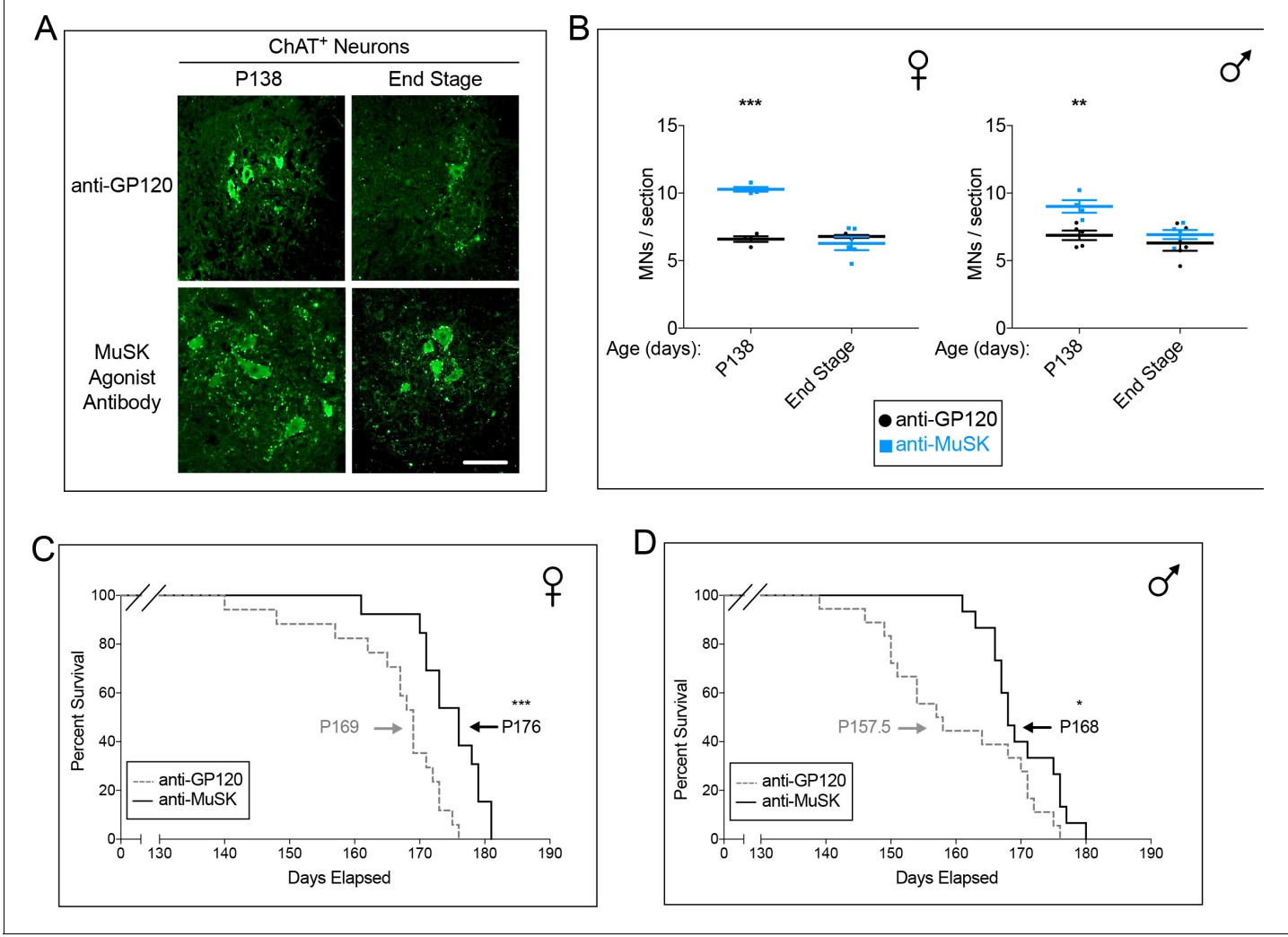

**Figure 6.** Chronic dosing with the MuSK agonist antibody increases motor neuron survival and extends the lifespan of *SOD1-G93A* mice. (**A**) Representative images of lumbar spinal cords stained with antibodies to ChAT (scale bar = 100 µm). (**B**) At P138, during the peak period of motor neuron cell death, the number of spinal motor neurons in the lumbar enlargement is greater in *SOD1-G93A* mice treated with the agonist antibody to MuSK (■) than in mice treated with the control antibody to GP120 (●). The scatter plot shows the values for individual mice (n = 3 to 5), as well as the mean values and SEM; *p<0.05, **p<0.01, ***p<0.001. (**C, D**) Female and male *SOD1-G93A* mice chronically injected with the control antibody to GP120 have a life span of 169 and 157.5 days, respectively (dotted line). Chronic injection of the reverse chimera MuSK agonist antibody prolongs longevity by 7 and 10 days in female and male *SOD1-G93A* mice, respectively (solid line). n ≥ 13; *p<0.05, **p<0.01, ***p<0.001.
DOI: https://doi.org/10.7554/eLife.34375.014

axons that have fully withdrawn, early delivery of a MuSK agonist is likely to be more effective than later delivery in ALS.

However, ALS is a diagnosis of exclusion, leading to delays in diagnosis. Nonetheless, even at late stages of disease, a majority of synapses in *SOD1-G93A* mice are partially innervated, and the MuSK agonist antibody improved nerve terminal coverage at these partially innervated synapses. These findings suggest that the MuSK agonist antibody may also be effective if introduced later during disease. However, because overt motor deficits become evident in *SOD1-G93A* mice only a month before death, this aggressive mouse model of ALS may not be the optimal and most informative model to infer whether later introduction of the MuSK agonist antibody can stabilize synapses and slow motor dysfunction in ALS.

The loss of motor neurons during embryonic development is regulated, at least in part by synapse formation (*Hollyday and Hamburger, 1976*; *Tanaka and Landmesser, 1986*; *Landmesser, 1992*).

The increased survival of motor neurons in MuSK agonist antibody-injected *SOD1-G93A* mice indicates that adult motor neurons can likewise receive trophic support from muscle. Thus, preserving neuromuscular synapses not only maintains the essential attachment of nerve to muscle but also provides the added benefit of promoting motor neuron survival.

Although we used an agonist antibody to MuSK to stimulate retrograde signaling from muscle, one can envisage other approaches to stimulate MuSK or enhance retrograde signaling in order to maintain attachment of motor axons to muscle. The MuSK agonist antibody is effective at maintaining neuromuscular synapses in *SOD1-G93A* mice up to P162, but within the next week, synapses are lost, and the mice die. Because the MuSK agonist antibody is designed to maintain neuromuscular synapses and does not directly target or address the underlying cause of the disease and other pathologies in *SOD1-G93A* mice and ALS, the benefit of increasing retrograde signaling from muscle to nerve and promoting nerve terminal attachment is limited. Nonetheless, although the antibody cannot override the many pathological pathways that occur in the motor neuron and in non-neuronal cells, this therapeutic approach has a potent effect on the course of disease, reducing synaptic loss, improving motor output and extending the lifespan of *SOD1-G93A* mice longer than riluzole, the long-standing FDA approved treatment for ALS (*Jablonski et al., 2014*). Motor neuron cell death is a critical feature in ALS, but elimination of Bax, which prevents apoptotic cell death, fails to preserve neuromuscular synapses and increases survival of *SOD1-G93A* mice by only 20 days (*Gould et al., 2006*). Together with our studies, these findings give credence to the idea that combinatorial therapeutic interventions, including those that preserve neuromuscular synapses, will be necessary to fully address the complex pathology and symptoms of ALS and contribute to an improved quality of life for patient and family.

# Materials and methods

## Key resources table

| Reagent type (species) or resource | Designation | Source or reference | Identifiers | Additional information |
|---|---|---|---|---|
| strain, strain background (mouse) | Mouse: B6.Cg-Tg(SOD1*G93A)1Gur/J | The Jackson Laboratory | RRID:IMSR_JAX:004435 | |
| strain, strain background (mouse) | Mouse: C57BL/6J | The Jackson Laboratory | RRID:IMSR_JAX000664 | |
| cell line (mouse) | Mouse C2C12 skeletal muscle cells | Burden lab | ATCC Cat# CRL-1772, RRID:CVCL_0188 | |
| cell line (mouse) | Mouse: Immortalized wild type muscle cells | Burden lab | PMID: 18848351 | |
| cell line (mouse) | Mouse: Immortalized *Lrp4* mutant muscle cells | Burden lab | PMID: 18848351 | |
| antibody | Rabbit anti-Neurofilament-L | SYnaptic SYstems | Cat# 171 002, RRID:AB_887743 | |
| antibody | Rabbit anti-Synapsin 1/2 | SYnaptic SYstems | Cat# 106 002, RRID:AB_887804 | |
| antibody | Alexa 647-anti-human IgG | Life Technologies | Cat# A-21445, RRID: AB_2535862 | |
| antibody | anti-Choline Acetyltransferase | Millipore | Cat# AB144P-200UL, RRID:AB_90661 | |
| antibody | anti-NeuN | Millipore | Cat# MAB377, RRID: AB_2298772 | |
| antibody | Rabbit anti-MuSK | Burden lab | PMID: 10781064 | |
| antibody | Goat anti-MuSK | R and D Systems | Cat# AF3904, RRID:AB_2147242 | |
| antibody | Rabbit anti-Dok-7 | Burden lab | PMID: 18848351 | |
| antibody | Mouse anti-phosphotyrosine 4G10 | Millipore | | |
| peptide, recombinant protein | Alexa 594-alpha-bungarotoxin | Life Technologies | Cat#B13423 | |
| peptide, recombinant protein | Alexa 488-alpha-bungarotoxin | Life Technologies | Cat#B13422 | |

*Continued on next page*

*Continued*

| Reagent type (species) or resource | Designation | Source or reference | Identifiers | Additional information |
|---|---|---|---|---|
| peptide, recombinant protein | Hoechst 33342 | Thermo Fisher | Cat# 62249 | |
| peptide, recombinant protein | Recombinant Rat Agrin Protein | R and D Systems | Cat#550-AG-100 | |
| commercial assay or kit | GAPDH TaqMan Assay Mm00186822_cn | Thermo Fisher | Cat# 4400291 | |
| software, algorithm | Prism 7.0 | http://www.graphpad.com/scientific-software/prism/ | RRID:SCR_002798 | |
| software, algorithm | Volocity 3D Image Analysis Software | http://www.perkinelmer.com/pages/020/cellularimaging/products/volocity.xhtml | RRID:SCR_002668 | |
| software, algorithm | pCLAMP Software Suite | https://www.moleculardevices.com/systems/axon-conventional-patch-clamp/pclamp-11-software-suite | | |
| other | DietGel 76A | ClearH20 | Cat#72-07-5022 | |

## Study design

The investigators were blinded from knowing whether mice were treated with the MuSK agonist or control antibody while acquiring and initially analyzing data. Data are presented as mean ±SEM. Statistical comparisons between groups were analyzed using an unpaired, two-tailed Student's $t$-test, log-rank test (survival), linear regression (CMAPs), or two-way ANOVA (failures). Statistical analyses were conducted using Prism 7.0 software (GraphPad Software). The number (n) of mice used to calculate the mean, SEM values and the confidence limits (p values) are indicated in the figure legends.

## Mice

The copy number of the human *SOD1-G93A* gene was routinely quantified by TaqMan real-time PCR and normalized to GAPDH (Life Technologies Assay# Mm00186822_cn). All mice included in this study had 21–26 copies of *hSOD1-G93A* (*Figure 3—figure supplement 1*). DietGel 76A (ClearH$_2$0) was placed on the cage floor so that mice had ready access to nourishment. Others have measured the lifespan *SOD1-G93A* mice by placing mice on their side and sacrificing mice if they were unable to right themselves in 15 s. Because we were concerned that this assay reported on limb muscle function and may not be temporally aligned with the time of death, we used a variant assay, which provided an accurate measure of longevity. When mice were unable to right themselves to eat or drink over the course of several hours, they invariably succumbed within a day; we defined this time as disease end-point and sacrificed mice at this time. Mice were housed and maintained according to Institutional Animal Use and Care Committee (IACUC) guidelines.

## Histology

Diaphragm muscles were stained with Alexa 594-conjugated α-bungarotoxin (α-BGT) (Life Technologies, Carlsbad, CA) to mark AChRs and rabbit antibodies to Neurofilament-L (SYnaptic Systems, Goettingen, Germany) and Synapsin 1/2 (SYnaptic Systems, Goettingen, Germany) to label axons and nerve terminals, as described previously (*Jaworski and Burden, 2006*; *Friese et al., 2007*). At fully innervated synapses, nerve terminal staining completely overlapped with postsynaptic AChRs, whereas nerve terminals were absent from original synaptic sites, marked by AChRs, at fully denervated synapses. At partially innervated synapses, nerve terminals occupied only a portion of the postsynaptic membrane. We examined a minimum of 50 synapses in the diaphragm muscle from each mouse and designated each synapse as fully innervated, partially innervated, or fully denervated. At each partially innervated synapse, the percentage of AChR-stained postsynaptic membrane that was apposed by Synapsin-stained nerve terminals was quantified using Volocity imaging software (PerkinElmer, Waltham, MA). To visualize and quantify staining of the agonist antibody, containing human Fc, at the neuromuscular junction, we used an Alexa 647-conjugated anti-human secondary (Life Technologies, Carlsbad, CA). Whole mounts of muscles were imaged with a Zeiss

LSM800 confocal microscope, and the fluorescent signal was quantified as described previously (*Jaworski and Burden, 2006*; *Friese et al., 2007*).

Spinal cords were dissected from mice perfused with 4% formaldehyde. Frozen sections (20 µm) of the lumbar region were stained with antibodies to choline acetyltransferase (ChAT) (AB144P-200UL from Millipore, Billerica, MA). We defined motor neurons as cells in the ventral horn of the lumbar spinal cord that were positive for ChAT, excluding ChAT-positive preganglionic and Pitx2-positive neurons. We only counted ChAT-stained cells with a clearly defined nucleus in order to avoid double-counting motor neurons in multiple sections. We analyzed ~10 sections, evenly spaced in the lumbar enlargement, which together contained >50 motor neurons in each mouse.

## Antibody-binding and MuSK phosphorylation

Chimeric antibodies were produced by transferring cDNAs encoding the variable regions of MuSK agonist antibody #13 to expression vectors containing the mouse kappa and IgG2a constant region. MuSK agonist antibodies were produced in CHO cells and purified by Protein A and size exclusion chromatography. The activity of the reverse chimera antibody for stimulating clustering of AChRs in C2 myotubes was similar to that for the human agonist antibody to MuSK (*Figure 4—figure supplement 1*). Fab fragments were prepared by protease digestion of human IgG1 followed by removal of uncleaved IgG and Fc fragments on a Protein A Sepharose column and size exclusion chromatography.

We used a solid-phase binding assay to measure binding between the MuSK agonist antibody and the extracellular (ecto) region (E22 to T494), the first three Ig-like domains (E22 to I103) or the Frizzled-like domain (D312 to K456) from mouse MuSK (*Zhang et al., 2011*). Maxisorp plates were coated with MuSK agonist antibody #13 (5µg/ml), and subsequently incubated with 8-His-tagged MuSK proteins, followed by a horseradish peroxidase (HRP) conjugated antibody to 8-His. Bound HRP was quantified by measuring HRP activity (Thermo scientific#34028).

C2C12 muscle cells were purchased from the ATCC and were not tested for mycoplasma prior to use. These C2C12 muscle cells, as well as immortalized wild type or *Lrp4* mutant muscle cells, were differentiated and treated with either neural Agrin (1 nM) or antibodies (10 nM). MuSK was immuno-precipitated from lysates, and MuSK expression and MuSK tyrosine phosphorylation were measured by probing Western blots, as described previously (*Herbst and Burden, 2000*). C2C12 cells were grown in 24-well culture plates in DMEM with 10% fetal bovine serum (FBS) until myoblasts were 70% confluent. Myoblasts were then allowed to differentiate into myotubes by replacing the FBS with 2% horse serum. After 7 days, the cultures were treated for 16 hr with varying concentrations of rcMuSK antibody #13 or a Fab from antibody MuSK #13. Cells were fixed in 4% paraformaldehyde and stained with Alexa 488 conjugated- α-BGT. Two to four images were collected from each well, and the number of AChR clusters was analyzed using imageJ software. Neural Agrin (10 nM) (R and D Systems, Minneapolis, MN) was used as a positive control for AChR clustering (data not shown).

Hind-limb muscles were denervated by cutting the sciatic nerve, as described previously (*Simon et al., 1992*). Four days after denervation, mice were injected with MuSK agonist antibody #13, and we measured MuSK expression and MuSK tyrosine phosphorylation 3 days later. MuSK and Dok-7 were immunoprecipitated from lysates, and their expression levels were determined by Western blotting (*Herbst and Burden, 2000*; *Hallock et al., 2010*). MuSK tyrosine phosphorylation was measured by probing Western blots with antibody 4G10, as described previously (*Herbst and Burden, 2000*; *Hallock et al., 2010*).

## Recording and evaluation of compound muscle action potentials (CMAPs) from the diaphragm muscle

To assess the function of neuromuscular junction in the mouse diaphragm muscle (*Figure 5—figure supplement 1*), we developed an ex vivo phrenic nerve-diaphragm preparation. We studied the diaphragm muscle from ~P140 male and ~P150 female mice, which is three to four weeks prior to end-stage, respectively. We did not use the in vivo preparation, described by others (*Lepore et al., 2011*), because we were concerned that in vivo stimulation of the phrenic nerve, at moderate to high frequencies, would lead to variable and unreliable CMAP recordings, likely due to changes in the electrode position caused by muscle contraction. Moreover, a related method, reported to record from the mouse diaphragm muscle, uses a surface recording electrode, and likely monitors

the activity of multiple thoracic muscles (*Martin et al., 2015*). Thus, following anesthesia with 5% iso-flurane, mice were decapitated, and the diaphragm muscle, together with the phrenic nerve, was quickly isolated and transferred to a customized recording chamber. The chamber was perfused continuously with oxygenated (95% $O_2$ and 5% $CO_2$) artificial cerebrospinal fluid solution (128.25 mM NaCl, 4 mM KCl, 0.58 mM $NaH_2PO_4$, 21 mM $NaHCO_3$, 30 mM D-glucose, 1.5 mM $CaCl_2$, and 1 mM $MgSO_4$) at a rate of ~10 ml/min at room temperature (~20–24°C). The phrenic nerve that innervates the left hemi-diaphragm muscle was stimulated by drawing the distal part of the left phrenic nerve into a suction electrode (*Figure 5—figure supplement 1*). We validated proper positioning of the stimulating electrode by visually inspecting muscle contractions following stimulation of the phrenic nerve. EMG activity was recorded using a suction electrode placed in the upper left quadrant of the muscle, 1 mm toward the costal side of the main intramuscular nerve and endplate zone in the middle of the muscle. A light suction was applied to the recording electrode to secure a tight seal between the tip of the electrode and the muscle fibers. In this manner, damage to the diaphragm muscle was avoided, which was confirmed by observing muscle contractions during stimulation. The phrenic nerve was stimulated with square pulses (0.2 ms in duration) at several frequencies (1 Hz to 50 Hz) for 60 s. The intensity of stimulation was progressively increased from the threshold, defined as the minimum response in three out of five trials, until the CMAP reached a maximal response.

The stimulation intensity was set at twice the intensity required for the maximal response to ensure a supra-maximal intensity of stimulation (25μA to 200μA). Recordings were accepted for analysis only when the CMAP amplitude (peak-to-peak) was unchanged following 1 Hz stimulation. The amplitudes of the evoked CMAPs at higher frequencies were expressed as a percentage of the first evoked CMAP for the entire duration of stimulation. Recordings were fed to an A/D interface (Digidata 1440A, Molecular Devices) and acquired with Clampex (v10.2, Molecular Devices) at a sampling rate of 50 kHz. Data were analyzed off-line using Clampfit (v10.2, Molecular Devices). We defined CMAP failures as the absence of an evoked response discernable from the background noise recorded prior to the stimulation.

## Acknowledgements

We are grateful to Martin Raff, Ruth Lehmann and Maartje Huijbers for their comments on the manuscript. We thank Richard Scheller and Jagath Junutula at Genentech for their assistance and commitment in resuscitating the agonist antibodies described here. We thank Joseph Lewcock, Gai Ayalon, Arundhati Sengupta Ghosh and Isidro Hotzel at Genentech for providing the MuSK agonist antibodies as well as the control antibodies to ragweed pollen and GP120 and for in vitro data. We thank Elena Michaels for her assistance with confocal microscopy and colony maintenance, and Yonglei Shang for his assistance with the antibody-binding assays. We are grateful to Fernando Vieira and Valerie Tassinari at ALSTDI for their assistance with the qPCR assay for measuring *SOD1-G93A* copy number. This work was supported with funds from the NIH (R37 NS36193), the Robert Packard Center for ALS Research, the ALS Association, and Above and Beyond LLC to SJB and with support from the NIH (RO1NS078375) to GZM. S.C is grateful for support from the Molecular, Cellular, and Translational Neuroscience Training Grant (NIH NINDS, T32 NS86750).

## Additional information

### Competing interests

Steven J Burden: holds a patent (#9,329,182) for 'Method of treating motor neuron disease with an antibody that agonizes MuSK'. The other authors declare that no competing interests exist.

### Funding

| Funder | Grant reference number | Author |
| --- | --- | --- |
| ALS Association | | Steven J Burden |
| National Institute of Neurological Disorders and Stroke | R37 NS36193 | Steven J Burden |

| National Institute of Neurological Disorders and Stroke | RO1 NS078375 | George Z Mentis |
| National Institute of Neurological Disorders and Stroke | T32 NS86750 | Sarah Cantor |

The funders had no role in study design, data collection and interpretation, or the decision to submit the work for publication.

## Author contributions

Sarah Cantor, Conceptualization, Data curation, Formal analysis, Validation, Investigation, Visualization, Methodology, Writing—original draft, Writing—review and editing; Wei Zhang, Conceptualization, Data curation, Formal analysis, Validation, Investigation, Methodology, Writing—original draft, Writing—review and editing; Nicolas Delestrée, Conceptualization, Resources, Data curation, Formal analysis, Validation, Investigation, Visualization, Methodology, Writing—original draft, Writing—review and editing; Leonor Remédio, Conceptualization, Resources, Data curation, Formal analysis, Supervision, Funding acquisition, Validation, Investigation, Visualization, Methodology, Writing—original draft, Writing—review and editing; George Z Mentis, Steven J Burden, Conceptualization, Resources, Data curation, Formal analysis, Supervision, Funding acquisition, Validation, Investigation, Visualization, Methodology, Writing—original draft, Project administration, Writing—review and editing

## Author ORCIDs

Sarah Cantor http://orcid.org/0000-0002-8675-3729
Leonor Remédio http://orcid.org/0000-0002-1509-0024
Steven J Burden http://orcid.org/0000-0002-3550-6891

## Ethics

Animal experimentation: All procedures were approved and mice were maintained according to Institutional Animal Use and Care Committee (IACUC protocol number 160425) guidelines at NYU Medical School.

## Decision letter and Author response

Decision letter https://doi.org/10.7554/eLife.34375.017
Author response https://doi.org/10.7554/eLife.34375.018

## Additional files

### Supplementary files

• Transparent reporting form
DOI: https://doi.org/10.7554/eLife.34375.015

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
