## [Decision Letter]

Thank you for submitting your article "Preserving Neuromuscular Synapses in ALS by Stimulating MuSK with a Therapeutic Agonist Antibody" for consideration by *eLife*. Your article has been favorably evaluated by Didier Stainier (Senior Editor) and five reviewers, one of whom, Jonathan Glass (Reviewer #4), is a member of our Board of Reviewing Editors. The following individuals involved in review of your submission have agreed to reveal their identity: Mark Rich (Reviewer #1); Victor Faundez (Reviewer #3).

The reviewers have discussed the reviews with one another and the Reviewing Editor has drafted this decision to help you prepare a revised submission.

Summary:

Denervation of skeletal muscle is an early and prominent pathological feature of animal models as well as human ALS. Here, Cantor, et al., extend Dr. Burden's previous publication demonstrating that genetic manipulation of MuSK expression reduces progressive denervation in the *G93A SOD1* mouse model of ALS. The authors show that intraperitoneal administration of an agonist antibody to MuSK started at symptom onset (P90) delays denervation of the diaphragm, decreases motor neuron loss in the lumbar spinal cord, and modestly prolongs survival. The data are well-presented and compelling, providing nice proof of principle for this mechanism of action for further therapy development as a pharmacological approach to prevention of denervation in ALS and other denervating diseases. Furthermore, this intervention is agnostic to the underlying cause of denervation, and may be applicable to both genetic and non-genetic causes of ALS.

The reviewers were generally positive about the data presented, but raised some concerns. The importance of improving survival by only several days in this animal model was questioned. You mention in your final paragraph in the Discussion that "this antibody cannot override the many pathological pathways that occur in the motor neuron and other non-neuronal cells", and that this would represent an addition to a "combinatorial" intervention. You might expand on this message by referring to the models that show little effect on survival when motor neuron cell bodies alone are protected without protection of NMJs.

There was also concern about the translational potential of this approach using repeated injections of an antibody. Specifically, there was a concern that repeated injections would generate an immune response to the antibody, causing a myasthenia-like disorder or even accelerated denervation. Please address this issue in your revised manuscript.

Essential revisions:

1) In the Introduction, the authors state "Although defects in the MuSK signaling pathway are not associated with ALS". However, it was recently reported that *SOD1-G93A* myofibers show reduced AChR clustering associated with markedly reduced MuSK and phospho-MuSK levels in vitro and in vivo (120 days, hind-limb muscles, Vilmont et al., 2016, Scientific Reports, doi: 10.1038/srep27804. It is not determined by Vilmont and colleagues whether the drop in MuSK is a cause or a consequence of denervation, nevertheless this may be a major factor in the limitation of MuSK agonist therapy as animals approach end-stage. The authors should discuss this and consider showing levels of MuSK and phospho-MuSK over time in treated vs. untreated animals if muscle tissue is still available from treated mice.

2) The authors initially present pharmacokinetic data using a human antibody, and then transition to a mouse reverse chimera for chronic dosing. The human PK data is used to guide the dosing strategy. While the potency of these antibodies is identical for inducing AChR clustering in vitro, the blood levels of mouse antibody achieved at 10 mg/ml are only in the range of 1-2 µg/ml (Figure 4—figure supplement 1), whereas the human antibody produced initial levels from 50-100 µg/ml (Figure 2). If this is the case, the authors need to show NMJ saturation levels and validate MuSK phosphorylation in vivo with the mouse dosing paradigm, as these levels are considerably lower.

3) The authors do not confirm that the reverse chimera antibody is triggering MuSK phosphorylation in the treated *SOD1-G93A* mice. If the blood levels achieved with the mouse antibody are really 50-100 times lower, it becomes quite important to validate that receptors are being saturated (straightforward immunostaining as was done in Figure 2, and phospho-MuSK vs. MuSK Western blot as was done in Figure 1—figure supplement 1).

4) Do the authors have any data on NMJ morphology or physiology in muscles other than diaphragm?

---

## [Author Response]

[…] The reviewers were generally positive about the data presented, but raised some concerns. The importance of improving survival by only several days in this animal model was questioned. You mention in your final paragraph in the Discussion that "this antibody cannot override the many pathological pathways that occur in the motor neuron and other non-neuronal cells", and that this would represent an addition to a "combinatorial" intervention. You might expand on this message by referring to the models that show little effect on survival when motor neuron cell bodies alone are protected without protection of NMJs.

Our study shows that preserving neuromuscular synapses increases survival by 7 to 11 days in female and male *SOD1-G93A* mice, respectively. As suggested, we have added a statement and reference in the Discussion referring to models that show a similar effect on survival when motor neuron cell death is prevented without protecting neuromuscular synapses: “Motor neuron cell death is a critical feature in ALS, but elimination of Bax, which prevents apoptotic cell death, fails to preserve neuromuscular synapses and increases survival of *SOD1-G93A* mice by only 20 days (Gould et al., 2006). Together with our studies, these findings give credence to the idea that combinatorial therapeutic interventions, including those that preserve neuromuscular synapses, will be necessary to fully address the complex pathology and symptoms of ALS and contribute to an improved quality of life for patient and family.”

There was also concern about the translational potential of this approach using repeated injections of an antibody. Specifically, there was a concern that repeated injections would generate an immune response to the antibody, causing a myasthenia-like disorder or even accelerated denervation. Please address this issue in your revised manuscript.

Repeated injections of therapeutic antibodies are commonly used to treat a variety of diseases, including arthritis, multiple sclerosis, asthma, osteoporosis, cancers, Crohn’s disease, muscular dystrophy; see: https://en.wikipedia.org/wiki/List of therapeutic monoclonal antibodies). Moreover, it is possible to reduce a potential immune response to the variable regions of the injected antibodies by having multiple, differing agonist antibodies available.

Essential revisions:1) In the Introduction, the authors state "Although defects in the MuSK signaling pathway are not associated with ALS". However, it was recently reported that SOD1-G93A myofibers show reduced AChR clustering associated with markedly reduced MuSK and phospho-MuSK levels in vitro and in vivo (120 days, hind-limb muscles, Vilmont et al., 2016, Scientific Reports, doi: 10.1038/srep27804. It is not determined by Vilmont and colleagues whether the drop in MuSK is a cause or a consequence of denervation, nevertheless this may be a major factor in the limitation of MuSK agonist therapy as animals approach end-stage. The authors should discuss this and consider showing levels of MuSK and phospho-MuSK over time in treated vs. untreated animals if muscle tissue is still available from treated mice.

Vilmont et al. show that cultured myotubes from *SOD1-G93A* mice show numerous defects in muscle structure (not found in *SOD1-G93A* mice or reported in ALS) and an impaired response to Agrin. The authors describe the same defects in muscle structure and Agrin-responsiveness in dynein-deficient muscle cultures. Because decreasing dynein expression results in general defects in muscle structure, we believe that the link between dynein, mutant forms of *SOD1* and MuSK expression is not particularly strong.

2) The authors initially present pharmacokinetic data using a human antibody, and then transition to a mouse reverse chimera for chronic dosing. The human PK data is used to guide the dosing strategy. While the potency of these antibodies is identical for inducing AChR clustering in vitro, the blood levels of mouse antibody achieved at 10 mg/ml are only in the range of 1-2 µg/ml (Figure 4—figure supplement 1), whereas the human antibody produced initial levels from 50-100 µg/ml (Figure 2). If this is the case, the authors need to show NMJ saturation levels and validate MuSK phosphorylation in vivo with the mouse dosing paradigm, as these levels are considerably lower.

We inadvertently failed to indicate that the values displayed along the y-axis in Figure 4—figure supplement 1 are log_10_ values. We have corrected this error in the revised version by displaying log values along the y-axis: the corrected Figure 4—figure supplement 1 shows similar blood values for the agonist antibody in mice injected with either the human antibody or the reverse chimera.

3) The authors do not confirm that the reverse chimera antibody is triggering MuSK phosphorylation in the treated SOD1-G93A mice. If the blood levels achieved with the mouse antibody are really 50-100 times lower, it becomes quite important to validate that receptors are being saturated (straightforward immunostaining as was done in Figure 2, and phospho-MuSK vs. MuSK Western blot as was done in Figure 1—figure supplement 1).

See response to comment #2.

4) Do the authors have any data on NMJ morphology or physiology in muscles other than diaphragm?

We did stain limb muscles to show that the MuSK agonist antibody bound to synapses in limb muscles as well as the diaphragm muscle, but all of our histological and physiological experiments were carried-out in the diaphragm muscle.